# QuAC: Quality-Adaptive Activation for Degraded Image Understanding

## Abstract

Degraded image understanding remains a significant challenge in computer vision. To mitigate the domain shift between high-quality and low-quality image distributions, we propose an adaptation approach based on activation functions rather than adjusting convolutional parameters. First, inspired by physiological findings in the human visual system, we introduce *Quality-adaptive Activation* (QuAC), a novel concept that automatically adjusts neuron activations based on input image quality to enhance essential semantic representations. Second, we implement *Quality-adaptive meta-ACON* (Q-ACON), which incorporates hyperparameters learned from image quality assessment functions. Q-ACON is efficient, flexible, and plug-and-play. Extensive experiments demonstrate that it consistently improves the performance of various networks—including convolutional neural networks, transformers, and diffusion models—against challenging degradations across multiple vision tasks, such as semantic segmentation, object detection, image classification, and image restoration. Furthermore, QuAC integrates effectively with existing techniques like knowledge distillation and image restoration, and can be extended to other activation functions. The code will be released after peer review.

## 1 Introduction

Image understanding tasks, such as semantic segmentation and object detection, have achieved significant progress over the past decade due to advances in deep learning techniques (Minaee et al., 2021; Liu et al., 2024). For example, the *Segment Anything Model* (SAM) (Kirillov et al., 2023) demonstrates remarkable performance in general image segmentation and has inspired numerous extensions (Ke et al., 2024; Zhang et al., 2023). However, practical applications like autonomous driving (Ji et al., 2024; Sakaridis et al., 2021) often encounter complex degradations, causing significant performance drops in these models when processing low-quality images (Kim et al., 2022; Chen et al., 2024).

To enhance degraded image understanding, the most straightforward approach is to improve image quality through *super-resolution* (SR) (Liu et al., 2022) or restoration techniques (Li et al., 2022). However, these methods heavily depend on restoration performance and often fail on severely degraded images (Pei et al., 2019). Moreover, they do not fundamentally improve the robustness of segmentation models. Another effective strategy involves adapting pretrained models using adapters and data augmentation (Wang et al., 2022; Endo et al., 2023; Chen et al., 2024; Zhang et al., 2024). Nevertheless, excessive use of degraded images may cause catastrophic forgetting, potentially degrading performance on high-quality images or generalization to out-of-distribution samples (Zhong et al., 2023; Xu et al., 2023).

To mitigate the distribution shift between high-quality and low-quality images (Zhang et al., 2024), existing methods primarily focus on adjusting model parameters. However, when processing two images with similar semantics but different quality levels, a fixed convolution filter produces significantly different neuron activations (Figure 1a). This highlights the importance of activation functions, which directly control feature rectification and scaling but remain underexplored. Furthermore, physiological studies of the human visual system (HVS) reveal that visual neurons in the cortex (Hubel & Wiesel, 1962; Kersten et al., 2004; Reynolds et al., 1999) actively suppress noise and ambiguous signals through feedback mechanisms, which can be modeled using dynamic

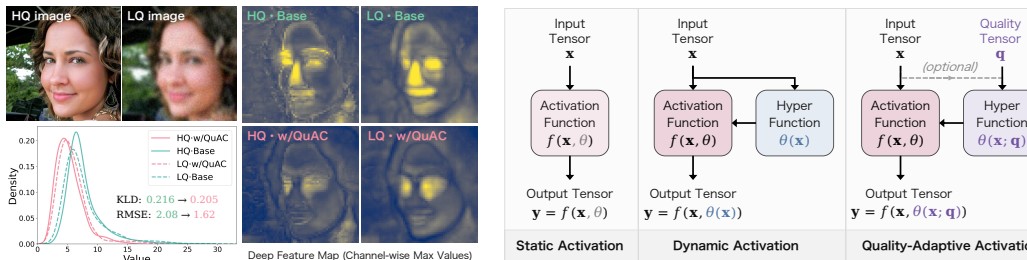

(a) Domain shift between HQ and LQ images.      (b) Comparison between QuAC and existing activations.

Figure 1: **Motivations.** (a) Activation distributions differ significantly between high-quality (HQ) images and their low-quality (LQ) counterparts in the base model (Guo et al., 2022). QuAC mitigates this divergence by reducing the Kullback–Leibler divergence (KLD) and root mean squared error (RMSE) between HQ and LQ activation distributions. (b) Comparison between *quality-adaptive activation* (QuAC) and existing static or dynamic activation functions.

thresholds (Gollisch & Meister, 2008). These findings inspire us to ask: *Can we develop learnable activation functions that adapt to input quality, enabling robust extraction of semantic features from degraded images?*

To this end, the main contributions of this paper are threefold:

*First*, we propose a novel concept of *Quality-adaptive Activation* (QuAC), inspired by dynamic activations (Chen et al., 2020; Ma et al., 2021). As summarized in Figure 1b, unlike static activations (e.g., ReLU (Jarrett et al., 2009)) that are identical for all inputs, or existing dynamic activations that derive parameters from the input tensor (Chen et al., 2020), our method learns activation parameters from a quality representation of the input. This "minor" change enables QuAC to enhance the capacity and robustness of deep neural networks for varying input quality.

*Second*, building on the QuAC concept, we develop an implementation based on meta-ACON (Ma et al., 2021) by modulating it with hyperparameters learned from image quality assessment (Wang & Bovik, 2006). The resulting *quality-adaptive meta-ACON* (Q-ACON) is efficient, flexible, and plug-and-play. The QuAC concept can be readily extended to other activation functions, such as ReLU Jarrett et al. (2009), demonstrating its broad applicability.

*Third*, we conduct extensive experiments across various vision tasks, including semantic segmentation (Chen et al., 2024), face parsing (Lee et al., 2020), object detection (Khanam & Hussain, 2024), image classification (He et al., 2016a), and image restoration (Zheng et al., 2024). Experimental results show that Q-ACON consistently improves the performance of diverse networks—including RobustSAM (Chen et al., 2024), YOLOv11 (Khanam & Hussain, 2024), and SinSR (Wang et al., 2024a)—against challenging degradations. Moreover, QuAC integrates effectively with knowledge distillation and image restoration techniques (Zhou et al., 2022), further enhancing degraded image understanding.

## 2 RELATED WORKS

**Activation Functions.** Activation functions are essential components in neural networks, introducing nonlinearity to enable complex function modeling. Early activation functions such as Sigmoid (Rumelhart et al., 1986) and Tanh (LeCun et al., 1989) are smooth and bounded but often suffer from vanishing gradients (Bengio et al., 1994). ReLU (Jarrett et al., 2009; Nair & Hinton, 2010) and its parametric variants (Maas et al., 2013; Clevert et al., 2016; Hendrycks & Gimpel, 2016) have gained popularity due to their efficiency and favorable properties. Swish (Ramachandran et al., 2017) explores optimal functions through neural architecture search (NAS), while other functions like Softplus (Dugas et al., 2000), Maxout (Goodfellow et al., 2013), and SiLU (Elfwing et al., 2018) also demonstrate performance improvements.

Recent research has focused on *dynamic activation* functions that learn input-adaptive hyperparameters (Chen et al., 2020). For example, DY-ReLU (Chen et al., 2020) generates piecewise functions

by learning parameters from input elements. The ACON family (Ma et al., 2021), particularly meta-ACON, can automatically switch between linear and nonlinear operations. More recent works include DiTAC (Chelly et al., 2024) and AdaShift (Cai, 2024), which adapt activation curves to changing data distributions, and Swish-T (Seo et al., 2024) and t-Sigmoid (Masoudian et al., 2024) that incorporate task-specific adjustments.

**Blind Image Quality Assessment (BIQA).** In the context of QuAC and image understanding, we utilize existing blind image quality assessment (BIQA) models to extract effective features representing image degradation (Wang & Bovik, 2006). Early BIQA methods primarily rely on hand-crafted features based on pixel value statistics (Mittal et al., 2012b;a) or transformed coefficients (Gao et al., 2013; Moorthy & Bovik, 2011). While computationally efficient, these approaches are typically effective only for limited degradation types. Recent BIQA methods are predominantly based on deep neural networks (Madhusudana et al., 2022; Qin et al., 2023; Ke et al., 2021; Shi et al., 2024), which offer improved effectiveness at the cost of higher computational complexity.

**Degraded Image Understanding.** Various efforts have been made to address the challenges of degraded image understanding (Gao et al., 2025), particularly in segmentation and object detection tasks (Rajagopalan et al., 2023; Endo et al., 2023). Some methods aim to learn quality-agnostic features (Kim et al., 2021) or employ auxiliary classification guidance (Son et al., 2020). To adapt pretrained segmentation models to degraded images, existing approaches typically incorporate lightweight networks into the backbone to enhance representation capacity or reduce adaptation costs (Endo et al., 2023; Chen et al., 2024). Recent work by (Zhang et al., 2024) uses weakly supervised domain adaptation to improve SAM's performance on corrupted images. Additionally, many methods utilize knowledge distillation to align low-quality features with high-quality features (Feng et al., 2021; Wang et al., 2022; 2020; Chen et al., 2024). In this paper, we demonstrate that QuAC effectively cooperates with existing techniques to achieve better performance.

## 3 METHOD

Figure 2 presents the pipeline of *Quality-adaptive Activation* (QuAC) for image segmentation. We first define QuAC and then introduce its implementation through *quality-adaptive meta-ACON* (Q-ACON), discussing its application in image understanding tasks.

### 3.1 QUALITY-ADAPTIVE ACTIVATION (QUAC)

Following the framework of dynamic activation (Chen et al., 2020), we define *quality-adaptive activation* as follows. Given an input image $I$ encoded into a tensor $\mathbf{X}$ by a neural network, we use a quality function $\mathcal{F}_Q$ to obtain the quality representation:

$$\mathbf{q} = \mathcal{F}_Q(I). \tag{1}$$

The *quality-adaptive activation* is then defined as $f(\mathbf{X}, \theta(\mathbf{X}; \mathbf{q}))$, where the learnable parameters $\theta(\mathbf{X}; \mathbf{q})$ adapt to the quality and semantic (optional) properties of the input.

QuAC comprises three components:

- **Quality function** $\mathcal{F}_Q$: encodes the input sample into a quality representation $\mathbf{q}$.
- **Hyper function** $\theta(\mathbf{X}; \mathbf{q})$: computes activation parameters based on the input tensor $\mathbf{X}$ and quality tensor $\mathbf{q}$.
- **Activation function** $f(\mathbf{X}, \theta(\mathbf{X}; \mathbf{q}))$: generates activations for $\mathbf{X}$ using the parameters $\theta(\mathbf{X}; \mathbf{q})$.

In QuAC, the hyperparameters adapt to the quality of input sample $I$, enabling stronger representation capabilities for varying input quality compared to static or conventional dynamic activation functions.

### 3.2 IMPLEMENTATION OF QUAC

QuAC can be implemented by incorporating quality information into existing dynamic activation functions, as illustrated in Figure 1b. In this part, we take meta-ACON (Ma et al., 2021) as an example and extend it to *quality-adaptive meta-ACON* (Q-ACON).

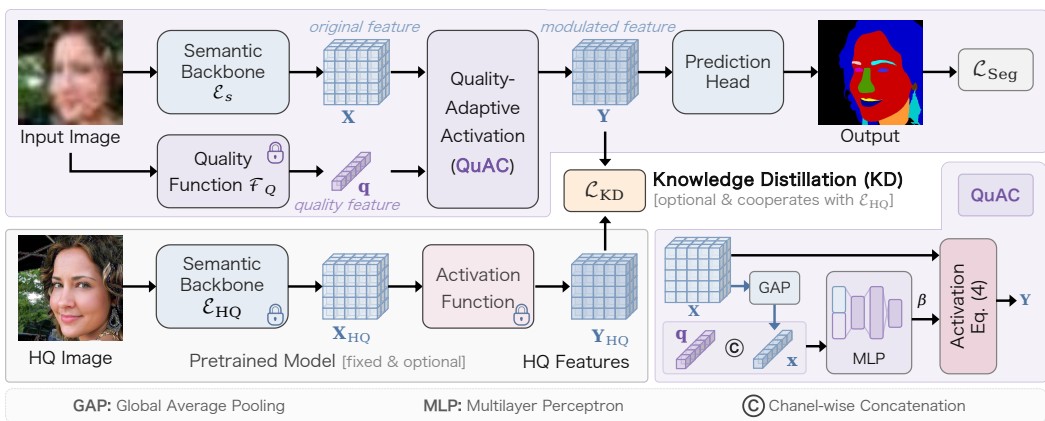

Figure 2: Pipeline of *Quality-Adaptive Activation* (QuAC) and its application. QuAC dynamically modulates semantic features $\mathbf{X}$ based on quality representation $\mathbf{q}$. Besides, the framework can optionally incorporate knowledge distillation (KD) for enhanced performance.

**Quality-adaptive meta-ACON (Q-ACON).** We implement QuAC by extending meta-ACON (Ma et al., 2021), a smooth, differentiable, and efficient activation function that achieves strong performance across various tasks. Following the QuAC concept, we learn the switching parameter $\beta$ based on both the input tensor $\mathbf{X}$ and quality tensor $\mathbf{q}$. The hyperfunction is formulated as:

$$\beta_Q = \mathrm{MLP}([\mathrm{GAP}(\mathbf{X}); \mathbf{q}]). \tag{2}$$

For an input $x \in \mathbf{X}$, Q-ACON outputs:

$$f(x) = (p_1 - p_2)\, x \cdot \sigma\left(\beta_Q \cdot (p_1 - p_2)\, x\right) + p_2 x. \tag{3}$$

Here, $p_1$ and $p_2$ are learnable channel-wise parameters from the original meta-ACON, determining the upper and lower bounds of its first derivative.

**Quality Function.** The quality function $\mathcal{F}_Q$ can utilize any quality assessment (QA) model that effectively represents signal quality in the target scenarios (Kim et al., 2022). In our implementation, we default to BRISQUE (Mittal et al., 2012a) for its balance of efficiency and effectiveness. BRISQUE extracts 36-dimensional handcrafted features based on spatial statistics of natural images, effectively characterizing various image degradations. We also demonstrate that different BIQA models consistently yield stable performance improvements (Section 4.5).

## 3.3 DISCUSSIONS

**(1) Minimum Design.** QuAC minimally modifies the hyperfunction to preserve the characteristics of original activation functions. For example, Q-ACON retains meta-ACON's smoothness, differentiability, and efficiency (Ma et al., 2021), while facilitating easy extension to other activation functions.

**(2) Flexible Structures.** Following meta-ACON (Ma et al., 2021), Q-ACON can be implemented as channel-wise, spatial-wise, or pixel-wise. We use the channel-wise structure recommended by meta-ACON to validate the QuAC concept, leaving other structures for future work.

**(3) Plug-and-Play Applications.** QuAC can be applied to various models and tasks by replacing original activation functions and incorporating a lightweight quality function. By default, we apply QuAC in the decoding stage for dense prediction tasks (Section 4.5).

**(4) Computational Efficiency.** The additional cost of QuAC depends primarily on the quality function. Using efficient quality representations (Kim et al., 2022; Mittal et al., 2012a), QuAC achieves stable performance with minimal overhead (Section 4.5).

### 3.4 Application to Image Understanding Tasks

Q-ACON can be applied to neural networks by replacing activation functions and incorporating a lightweight quality function. We use image segmentation as an example application and also introduce its optional integration with knowledge distillation.

**Example: Image Segmentation.** We apply Q-ACON in the decoding stage before the prediction head (Figure 2), as the encoder is more sensitive and replacing its activations may cause overfitting (Li et al., 2024; Ma et al., 2021). The training uses cross-entropy loss between predicted mask $\hat{\mathbf{M}}$ and ground-truth $\mathbf{M}$: $\mathcal{L}_{\text{Seg}} = \text{CE}(\hat{\mathbf{M}}, \mathbf{M})$.

**Cooperation with Knowledge Distillation (Optional).** Optionally, Q-ACON can integrate with knowledge distillation (KD) using a pretrained segmentation model $\mathcal{E}_{\text{HQ}}$ (Yang et al., 2022; Chen et al., 2024). As shown in Figure 2, we input the high-quality version of $I$ into $\mathcal{E}_{\text{HQ}}$ to obtain features $\mathbf{Y}_{\text{HQ}}$, then align Q-ACON's output $\mathbf{Y}$ with $\mathbf{Y}_{\text{HQ}}$ using Kullback–Leibler divergence: $\mathcal{L}_{\text{KD}} = \text{KLD}(\mathbf{Y}_{\text{HQ}}, \mathbf{Y})$. The total loss is: $\mathcal{L} = \lambda_1 \mathcal{L}_{\text{Seg}} + \lambda_2 \mathcal{L}_{\text{KD}}$. This KD guidance cooperates effectively with Q-ACON's flexibility to further enhance performance (Section 4.6).

## 4 Experiments

We evaluate the effectiveness and generalization of QuAC by integrating it into various deep learning models for multiple vision tasks, including image classification (He et al., 2016a), object detection (Khanam & Hussain, 2024; Hai et al., 2023), image segmentation (Chen et al., 2024; Guo et al., 2022), and image restoration (Zheng et al., 2024; Wang et al., 2024b; Zhou et al., 2024).

**Degradations.** To evaluate robustness against complex degradations, we adopt a hybrid degradation strategy commonly used in image restoration (Zhou et al., 2022). Given a high-quality image $I_h$, it is degraded as:

$$I_l = \{[(I_h \otimes k_\sigma)_{\downarrow_r} + n_\delta]_{\text{JPEG}_q}\}_{\uparrow_r}, \tag{4}$$

where $I_h$ undergoes Gaussian blurring with kernel $k_\sigma$, downsampling by factor $r$, random noise addition, JPEG compression, and finally upsampling to the original size. Following CodeFormer (Zhou et al., 2022), we construct five degradation levels (LQ$i$, $i = 1, \ldots, 5$) for task-specific datasets, with higher $i$ indicating more severe degradation. We also include real-world degradations from datasets like CODaN (Lengyel et al., 2021) and HazyDet (Feng et al., 2024), which contain naturally occurring low-quality images without additional processing.

### 4.1 Task I: Image Segmentation

**Settings.** We compare our approach with static activation functions (ReLU (Jarrett et al., 2009), GELU (Hendrycks & Gimpel, 2016), SiLU (Elfwing et al., 2018)) and dynamic activation functions (DY-ReLU-B (Chen et al., 2020), DiTAC (Chelly et al., 2024), meta-ACON (Ma et al., 2021)) on image segmentation tasks. We also implement a simplified Q-ACON variant (Q-ACON$_q$) that uses only the quality tensor $\mathbf{q}$ in the hyperfunction. Evaluation metrics include mean Intersection over Union (mIoU) and Dice Coefficient.

**Baselines.** We evaluate QuAC on two segmentation baselines: (1) *SegNeXt (Guo et al., 2022) for face parsing*: We apply hybrid degradations to CelebAMask-HQ (Lee et al., 2020), training on both pristine and augmented images, and evaluate on five degradation levels. (2) *RobustSAM (Chen et al., 2024) for semantic segmentation*: Following official settings, we use point-based prompts on the Robust-Seg dataset (Cheng et al., 2014) and apply hybrid degradations to MSRA10K test set to assess generalization. Note that DiTAC is excluded from RobustSAM due to its lack of multi-GPU support required for training.

**Face Parsing.** Table 1a presents performance across different hybrid degradation levels and the overall average. Q-ACON outperforms other activation functions, demonstrating that quality-adaptive activation enhances representation capabilities on degraded images. Notably, Q-ACON achieves an absolute improvement of over 1.27 points compared to meta-ACON (Ma et al., 2021). Even the simplified Q-ACON$_q$ variant, which uses only quality features, surpasses meta-ACON on heavily degraded images. Figure 3 further illustrates Q-ACON's superior face parsing results, particularly in detailed regions. These qualitative observations, consistent with quantitative results,

|  | Clear | | Degrade(seen) | | Average | |
|---|---|---|---|---|---|---|
|  | mIoU | Dice | mIoU | Dice | mIoU | Dice |
| ReLU | 71.79 | 81.78 | 59.68 | 71.16 | 61.70 | 72.93 |
| GELU | 72.69 | 82.65 | 59.84 | 71.33 | 61.98 | 73.22 |
| SiLU | 72.87 | 82.70 | 60.30 | 71.82 | 62.40 | 73.64 |
| DY-ReLU-B | 72.56 | 82.56 | 59.57 | 71.06 | 61.73 | 73.06 |
| DiTAC | 72.09 | 82.05 | 60.32 | 71.82 | 62.28 | 73.53 |
| meta-ACON | 72.72 | 82.59 | 60.04 | 71.52 | 62.16 | 73.37 |
| Q-ACON$_q$ | 72.76 | 82.58 | 60.40 | 71.89 | 62.46 | 73.67 |
| Q-ACON | **73.52** | **83.08** | **61.41** | **72.85** | **63.43** | **74.56** |

(a) SegNeXt on CelebAMask-HQ (Lee et al., 2020) with hybrid degradations (*seen*).

|  | Clear | | Degrade(seen) | | Average | |
|---|---|---|---|---|---|---|
|  | mIoU | Dice | mIoU | Dice | mIoU | Dice |
| ReLU | 89.46 | 93.94 | 85.99 | 91.75 | 86.20 | 91.89 |
| GELU | 89.56 | 93.95 | 86.44 | 92.02 | 86.64 | 92.14 |
| SiLU | 89.26 | 93.71 | 86.24 | 91.84 | 86.43 | 91.96 |
| DY-ReLU-B | 88.98 | 93.58 | 85.89 | 91.66 | 86.09 | 91.78 |
| DiTAC | - | - | - | - | - | - |
| meta-ACON | 89.50 | 93.90 | 86.11 | 91.78 | 86.32 | 91.91 |
| Q-ACON$_q$ | 90.02 | 94.26 | 86.92 | 92.38 | 87.12 | 92.49 |
| Q-ACON | **90.09** | **94.27** | **86.98** | **92.39** | **87.18** | **92.50** |

(b) RobustSAM performance on MSRA10K (Cheng et al., 2014) with 15 single-type degradations (*seen*).

|  | Clear | | Degrade(unseen) | | Average | |
|---|---|---|---|---|---|---|
|  | mIoU | Dice | mIoU | Dice | mIoU | Dice |
| ReLU | 89.46 | 93.94 | 76.22 | 85.08 | 78.43 | 86.56 |
| GELU | 89.56 | 93.95 | 77.58 | 86.14 | 79.58 | 87.44 |
| SiLU | 89.26 | 93.71 | 77.05 | 85.74 | 79.08 | 87.07 |
| DY-ReLU-B | 88.98 | 93.58 | 76.87 | 85.62 | 78.89 | 86.94 |
| meta-ACON | 89.50 | 93.90 | 77.01 | 85.71 | 79.10 | 87.07 |
| Q-ACON$_q$ | 90.02 | 94.26 | 77.31 | 85.94 | 79.43 | 87.33 |
| Q-ACON | **90.09** | **94.27** | **77.69** | **86.20** | **79.76** | **87.54** |

(c) RobustSAM performance on MSRA10K (Cheng et al., 2014) with hybrid degradations (*unseen-degradations / seen-data*).

|  | Clear | | Degrade(seen) | | Average | |
|---|---|---|---|---|---|---|
|  | mIoU | Dice | mIoU | Dice | mIoU | Dice |
| ReLU | 80.47 | 88.20 | 75.10 | 84.32 | 75.43 | 84.56 |
| GELU | 81.16 | 88.69 | 75.74 | 84.84 | 76.08 | 85.08 |
| SiLU | 80.65 | 88.24 | 75.46 | 84.56 | 75.78 | 84.79 |
| DY-ReLU-B | 80.42 | 88.18 | 75.11 | 84.40 | 75.44 | 84.64 |
| meta-ACON | 80.48 | 88.23 | 75.03 | 84.31 | 75.37 | 84.56 |
| Q-ACON$_q$ | **81.65** | **89.09** | **76.28** | **85.27** | **76.62** | **85.51** |
| Q-ACON | 81.21 | 88.75 | 76.01 | 85.06 | 76.34 | 85.29 |

(d) RobustSAM performance on NDD20 (Trotter et al., 2020), STREETS (Snyder & Do, 2019), and FSS-1000 (Li et al., 2020) (*unseen-data*).

Table 1: Image segmentation performance on diverse datasets, with *seen* or *unseen* degradations.

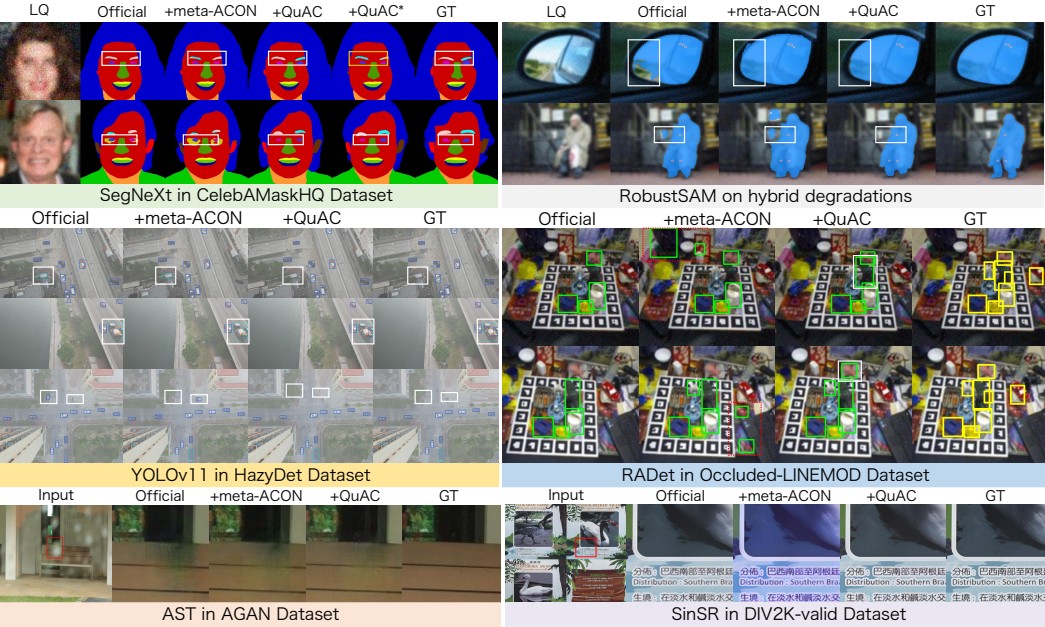

Figure 3: Qualitative evluation of QuAC on image segmentation, detection, and restoration tasks. * indicates the cooperation of QuAC with knowledge distillation.

demonstrate QuAC's effectiveness in enhancing segmentation model stability across varying input quality.

**Semantic Segmentation.** We evaluate RobustSAM on diverse datasets under four settings:

**(1) For the 15 seen distortion types on the MSRA10K test set** (Cheng et al., 2014). Q-ACON consistently improves segmentation accuracy on both clear and distorted images (Table 1b).

**(2) For the hybrid degradations on the MSRA10K test set (*unseen-degradations*),** Q-ACON also consistently improves performance, achieving competitive results on severely degraded images. Recall that the hybrid degradations are not used during training.

**(3) For the unseen datasets with 15 seen distortion types,** Q-ACON and Q-ACON$_q$ achieve the top-2 mIoU values on degraded images (Table 1d). Note that the NDD20 (Trotter et al., 2020), STREETS (Snyder & Do, 2019) and FSS-1000 (Li et al., 2020) datasets are not included in the training data. Accordingling, Figure 3 shows that Q-ACON improves the segmentation accuracy on degraded images in challenging scenarios. These observations solidly validate the significance of our core idea, QuAC, in boosting generalization capacity on out-of-distribution samples.

**(4) Generalization to real-world degradations (*unseen*).** To validate the capabilities against real-world degradations, we additionally evaluate the learned segmentation models on the BDD-100K (Yu et al., 2018), LIS (Chen et al., 2023), and ACDC (Sakaridis et al., 2021) datasets, which contain challenging degradations, including weather and lighting conditions. Additionally, we change the input prompt to box hints and replaced the quality function with CONTRIQUE (Madhusudana et al., 2022), to verify the performance of QuAC under different settings.

As shown in Table 2, Q-ACON leads to absolute improvements of 2.4 and 3.86 points on BDD100K and ACDC, respectively, but shows a decrease of 1.36 points on LIS. The possible reason might be the limited capacity of CONTRIQUE in representing dark images in LIS (Madhusudana et al., 2022). It is promising and necessary to boost the performance by exploring specific quality representations.

| RobustSAM | BDD100K | LIS | ACDC | | | | |
|---|---|---|---|---|---|---|---|
| *box hints* | (ALL) | (ALL) | (fog) | (night) | (rain) | (snow) | (AVG) |
| Official | 68.01 | **74.19** | 73.83 | 77.15 | 62.77 | 57.20 | 67.74 |
| meta-ACON | 63.66 | 73.13 | 64.50 | 77.75 | 50.45 | 47.16 | 59.97 |
| Q-ACON | **70.41** | 72.83 | **77.33** | **78.38** | **67.70** | **62.98** | **71.60** |

Table 2: Zero-shot segmentation performance (mIoU) for *real degradations (unseen)*, while using box hints.

## 4.2 TASK II: IMAGE CLASSIFICATION

**Settings.** We performed image classification on the CODaN (Lengyel et al., 2021) dataset using ResNet (He et al., 2016a) as the baseline model. CODaN is constructed from the high-quality ImageNet (Deng et al., 2009) and ExDark (Loh & Chan, 2019) datasets, and consists of 10,000 training images across 10 classes, along with 2,500 daytime and 2,500 nighttime test images. The model was trained for 100 epochs. To assess the effectiveness of different activation functions, we compared their top-k classification accuracy under consistent training conditions.

**Results.** As shown in Table 3a, Q-ACON improves the model's performance under different lighting conditions, especially at night, demonstrating its strong ability to adapt to varying degradations.

## 4.3 TASK III: OBJECT DETECTION

**Settings.** For the object detection task, we select two baselines to verify the effectiveness of our method. (1) We used YOLOv11 (Khanam & Hussain, 2024) as the base model and used the official settings for training. We compared the baseline model with two variants: one with meta-ACON and the other with Q-ACON. We used the HazyDet (Feng et al., 2024) dataset, which is designed for object detection in foggy conditions. (2) We select a challenging task, *i.e.*, occluded object detection, and conduct experiments on the Occluded-LINEMOD dataset (Brachmann et al., 2014). We adopt RADet (Hai et al., 2023) as the baseline and use its official experimental settings. The training and testing images are distorted with the five levels of hybrid degradation.

**Results.** As shown in Table 3b and Figure 3, the model incorporating Q-ACON demonstrates outstanding performance for degraded images in terms of *mean Average Precision* (mAP) with different thresholds (Lin et al., 2014; 2017), on both datasets.

## 4.4 TASK IV: IMAGE RESTORATION

**Settings.** We finally evaluate QuAC on image restoration tasks, including universal image restoration, image deraining and single image super-resolution reconstruction (SISR). For universal image restoration, we choose DiffUIR (Zheng et al., 2024) as the base model and add Q-ACON as a variant for comparison. We train and test according to the official settings of DiffUIR. For the deraining task, we use AST(Zhou et al., 2024) as the baseline network and add Q-ACON as a variant for comparison. We train and test AST and its variant with Q-ACON on the AGAN (Qian et al., 2018) dataset. For the SISR task, we use SinSR (Wang et al., 2024b) as the baseline model and add Q-

(a) **Image Classification.** ResNet (He et al., 2016a) on the CODaN dataset (Lengyel et al., 2021).

| | ResNet18 | | | | | | ResNet34 | | | | | | Average | | |
| | Day | | | Night | | | Day | | | Night | | | (Overall) | | |
| | Top1 | Top3 | Top5 | Top1 | Top3 | Top5 | Top1 | Top3 | Top5 | Top1 | Top3 | Top5 | Top1 | Top3 | Top5 |
|---|---|---|---|---|---|---|---|---|---|---|---|---|---|---|---|
| Official | 74.47 | 92.75 | 97.84 | 26.80 | 56.21 | 78.08 | 80.06 | 95.39 | 98.68 | 38.95 | 67.17 | 83.02 | 55.07 | 77.88 | 87.96 |
| + meta-ACON | 76.83 | 93.83 | 97.60 | 28.42 | 58.06 | 77.75 | 81.09 | 95.88 | 98.87 | 39.33 | 69.35 | 84.77 | 56.42 | 79.28 | 89.75 |
| + Q-ACON | 77.46 | 94.41 | 98.19 | 31.26 | 63.00 | 79.98 | 80.79 | 95.88 | 98.82 | 44.17 | 73.15 | 87.00 | 58.42 | 81.61 | 91.00 |

(b) **Object Detection**. Higher mAP indicates better detection accuracy.

| | YOLOv11 (on the HazyDet dataset) | | | | | | | | RADet (on the Occluded-LINEMOD dataset) | | | | | |
| | mAP | | | | mAP50 | | | | mAP | | | mAP75 | | |
| | Car | Track | Bus | Average | Car | Track | Bus | Average | Clear | Degrade | Average | Clear | Degrade | Average |
|---|---|---|---|---|---|---|---|---|---|---|---|---|---|---|
| Official | 53.80 | 20.30 | 53.70 | 42.60 | 80.80 | 32.30 | 73.40 | 62.10 | 63.20 | 41.48 | 45.10 | 74.70 | 46.58 | 51.27 |
| + meta-ACON | 53.50 | 20.30 | 53.50 | 42.50 | 80.80 | 32.50 | 73.60 | 62.30 | 63.50 | 42.24 | 45.78 | 75.00 | 47.66 | 52.22 |
| + Q-ACON | 53.80 | 20.90 | 53.40 | 42.70 | 81.00 | 33.60 | 74.00 | 62.80 | 63.60 | 42.40 | 45.93 | 75.10 | 48.26 | 52.73 |

(c) **Image restoration.** Higher PSNR/SSIM values indicate better clairity.

| | DiffUIR (Zheng et al., 2024) | | | | | | | | | | | | AST AGAN | | SinSR DIV2K-valid | |
| | Derain | | Enhancement | | Desnowing | | Dehazing | | Deblurring | | | | | | | |
| | PSNR↑ | SSIM↑ | PSNR↑ | SSIM↑ | PSNR↑ | SSIM↑ | PSNR↑ | SSIM↑ | PSNR↑ | SSIM↑ | PSNR↑ | SSIM↑ | PSNR↑ | SSIM↑ | PSNR↑ | SSIM↑ |
|---|---|---|---|---|---|---|---|---|---|---|---|---|---|---|---|---|
| Official | 28.40 | 0.8700 | 22.80 | 0.8783 | 28.87 | 0.8951 | 29.83 | 0.9518 | 27.27 | 0.8186 | | | 32.34 | 0.9355 | 26.28 | 0.7249 |
| + meta-ACON | 28.11 | 0.8648 | 22.38 | 0.8689 | 29.45 | 0.8968 | 30.88 | 0.9532 | 27.22 | 0.8166 | | | 32.21 | 0.9340 | 25.87 | 0.7436 |
| + Q-ACON | 28.16 | 0.8665 | 23.43 | 0.8841 | 30.01 | 0.9006 | 29.85 | 0.9490 | 27.51 | 0.8236 | | | 32.33 | 0.9364 | 27.70 | 0.7736 |

Table 3: Impacts of QuAC on image classification, object detection and image restoration tasks. In the table, **bold** represents the highest score, and underlined represents the second-highest score.

ACON to its output layer. Following settings of Real-ESRGAN (Wang et al., 2021), we fine-tune SinSR and its Q-ACON variant.

**Results.** As shown in Table 3c, Q-ACON enables competitive results on all metrics, confirming its effectiveness. Correspondingly, Figure 3 shows that the restored images with Q-ACON present clearer structures and fewer artifacts.

## 4.5 ANALYSIS OF QUAC

**Impact of Quality Functions.** We evaluate three Q-ACON variants using BRISQUE (Mittal et al., 2012a), Re-IQA (Saha et al., 2023), and CONTRIQUE (Madhusudana et al., 2022) as quality functions. Table 4 shows that CONTRIQUE and BRISQUE achieve better performance on severely distorted images. These results demonstrate that QuAC allows flexible choice of quality representations while consistently improving performance.

**Computational Cost.** Table 4 compares computational complexity. Q-ACON with BRISQUE achieves significant performance improvements with minimal parameter overhead—only about 2.3% over the ReLU baseline (27.57M parameters). This demonstrates QuAC's parameter efficiency and practical value in enhancing model expressiveness and generalization.

**Impact of Positions.** We analyze Q-ACON's placement by inserting it into the encoder, decoder, or both in SegNeXt. Table 5 shows that applying Q-ACON to encoder layers decreases performance, likely because early encoding layers are more sensitive and prone to overfitting (Li et al., 2024; Ma et al., 2021). Therefore, we apply Q-ACON after the backbone by default.

| SegNeXt | Q-ACON (mIoU) | | | Q-ACON + KD (mIoU) | | | Complexity | |
| | Clear | Degrade | avg. | Clear | Degrade | avg. | GFLOPs | Params (M) |
|---|---|---|---|---|---|---|---|---|
| BRISQUE | 72.22 | 60.26 | 62.25 | 74.25 | 61.73 | 63.82 | 32.51 | 28.21 |
| Re-IQA | 72.76 | 60.17 | 62.27 | 73.70 | 61.95 | 63.91 | 53.99 | 59.81 |
| CONTRIQUE | 72.44 | 60.10 | 62.16 | 73.95 | 62.32 | 64.26 | 32.52 | 59.82 |

| SegNeXt | mIoU | | |
| | Clear | Degrade | avg. |
|---|---|---|---|
| Encoder | 69.67 | 58.02 | 59.97 |
| Decoder | 72.44 | 60.10 | 62.16 |
| Both | 69.56 | 58.40 | 60.26 |

Table 4: Impact of quality functions & knowledge distillation (KD).   Table 5: Impact of positions.

## 4.6 EXTENSION EXPERIMENTS

**Cooperation with Knowledge Distillation.** We incorporate knowledge distillation (KD) into Seg-NeXt, as illustrated in Figure 2 and Section 3.4. As shown in Table 4, Q-ACON cooperates effectively with KD and further enhances performance. However, the first example in Figure 3 illustrates that the teacher network can sometimes mislead the student network (Stanton et al., 2021).

**Cooperation with Image Restoration.** We further analyze the cooperative performance of Q-ACON with image restoration. Specifically, we use CodeFormer (Zhou et al., 2022) to restore facial images and then apply the pre-trained face parsing models to them. As shown in Table 6, Q-ACON achieves the best overall performance, particularly on distorted images.

| SegNeXt | Clear | Degrade | avg. |
|---|---|---|---|
| ReLU (*off.*) | 71.79 | 57.59 | 61.62 |
| meta-ACON | 72.72 | 59.99 | 62.11 |
| Q-ACON | **73.52** | **62.79** | **64.58** |

Table 6: Cooperation with image restoration.

**Extension to Different Activation Functions: Quality-adaptive ReLU (Q-ReLU).** To validate the concept of *Quality-adaptive Activation* (QuAC), we extend it to the ReLU function, termed Q-ReLU. Following DY-ReLU-B (Chen et al., 2020), we concatenate the quality vector $\mathbf{q}$ with the input tensor $\mathbf{x}$ to learn the activation parameters $[a, b]$ (Figure 1b). For each neuron activation $x \in \mathbf{X}$, the output of Q-ReLU is defined as:

$$\text{Q-ReLU}(x) = a \cdot x + b \quad \text{with} \quad [a, b] = \text{MLP}(\text{GAP}(\mathbf{X}); \mathbf{q}), \tag{5}$$

where $[;]$ denotes channel-wise concatenation; GAP and MLP denote *global average pooling* and *multi-layer perceptron*, respectively. Similar to DY-ReLU (Chen et al., 2020), we implement a small, three-layer MLP with a reduction factor of $r = 16$ in the first layer, mapping the dimension back to the channel number. We integrate Q-ReLU into ResNet18 and evaluate it on the CODaN dataset. As shown in

| ResNet18 | Day | | | Night | | |
|---|---|---|---|---|---|---|
| | Top1 | Top3 | Top5 | Top1 | Top3 | Top5 |
| DY-ReLU | 76.09 | **94.02** | 97.89 | 32.50 | 63.05 | 80.08 |
| Q-ReLU | **76.19** | 93.97 | **98.38** | **35.48** | **64.33** | **82.16** |

Table 7: Quantitative evaluation of Q-ReLU.

Table 7, Q-ReLU improves the model's classification performance under nighttime conditions.

## 5 CONCLUSIONS

In this work, we address the challenge of degraded image understanding by proposing a novel concept of *quality-adaptive activation* (QuAC) and its implementation, Q-ACON. Extensive experiments demonstrate that QuAC effectively enhances various networks—including convolutional neural networks (*e.g.*, SegNeXt), transformers (*e.g.*, RobustSAM), and diffusion models (*e.g.*, SinSR)—across diverse image understanding and low-level vision tasks. In the future, we will explore the potential of this quality-adaptation concept for network design and additional tasks toward building universal models.

## 6 ETHICS STATEMENT

This work does not involve human subjects or sensitive personal data. All datasets are publicly available and used in compliance with their licenses. Our methods are intended solely for research purposes, and we will release code and documentation to support transparency and reproducibility.

## 7 REPRODUCIBILITY STATEMENT

We have made efforts to ensure the reproducibility of our results. The model architecture, training settings, and evaluation metrics are described in detail in the main paper. Additional implementation details, data preprocessing steps, and experimental settings are provided in the appendix and supplementary materials.

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

# A  APPENDIX

## A.1  PLUG-AND-PLAY APPLICATIONS OF QUAC

To validate the effectiveness of QuAC, we conducted extensive experiments on multiple low-quality image tasks. By default, we apply QuAC in the latter stage of the entire network, *i.e.*, the decoder stage. For the image classification task, we chooose ResNet He et al. (2016b) as the baseline. For the image detection task, we choose YOLOv11 Khanam & Hussain (2024) and RADetHai et al. (2023) as the baseline. For the face parsing experiment, we choose SegNeXt Guo et al. (2022) as the baseline. For the general image segmentation task, we choose RobustSAM Chen et al. (2024) as the baseline. For the image deraining task, we use AST Zhou et al. (2024) as the baseline. For the single-image super-resolution (SISR) task, we choose SinSR Wang et al. (2024b) as baseline. For each base network, we apply QuAC in the following settings:

- **ResNet.** We add QuAC after the fourth convolutional block and before the average pooling layer.

- **SegNeXt.** We add QuAC after Hamburger in the SegNeXt decoder Guo et al. (2022), that is, before the multilayer perceptron (MLP) for mask prediction.

- **RobustSAM.** In RobustSAM Chen et al. (2024), we add QuAC at the end of *Anti-degradation Mask Feature Generation* (AMFG) in MaskDecoder, before the feature fusion stage for mask prediction.

- **YOLOv11**. We also insert QuAC into the C3k2 module block.

- **RADet.** We choose RADetHai et al. (2023) as baseline, Q-ACON is added to the end of decoder head.

- **AST.** For image deraining, we use ASTZhou et al. (2024) as the baseline network and add three Q-ACONs at the end of the encoder, the bottleneck, and the beginning of the decoder.

- **SinSR.** For the SISR task, we adopt SinSR Wang et al. (2024b) as baseline networks and integrate the proposed Q-ACON activation into their output layers.

## A.2  HYBRID DEGRADATION SETTINGS

To simulate complex degradation, we augment every original high-quality (HQ) image $I_h$ using ***hybrid degradation*** following advanced image restoration works Zhou et al. (2022), which is formulated as:

$$I_l = \{[(I_h \otimes k_\sigma)_{\downarrow_r} + n_\delta]_{\text{JPEG}_q}\}_{\uparrow_r}, \tag{6}$$

$I_h$ is first blurred with a Gaussian kernel $k_\sigma$, then downsampled by a factor of $n$, followed by additional random noise and JPEG compression, and finally upsampled to its original size. Following CoderFormer Zhou et al. (2022), we set 5 levels of hybrid degradations, by randomly choose the degradation parameters in separate intervals, as shown in Table 8. The augmented images are denoted by $LQi, i = 1, ...5$ in the following part; a greater value of $i$ generally indicates heavier degradation.

| Degradation: $I_l = \left\{[(I_h \otimes k_\sigma) \downarrow_r + n_\delta]_{\text{JPEG}_q}\right\} \uparrow_r$ | | | | | |
|:---:|:---:|:---:|:---:|:---:|:---:|
| Level | $k_\sigma$ | $\sigma$ | $r$ | $\delta$ | $q$ |
| LQ1 | 41 | [1.0, 2.4] | [1.5, 1.7] | [0, 5] | [85, 90] |
| LQ2 | 41 | [2.5, 4.9] | [1.8, 2.0] | [5, 10] | [75, 80] |
| LQ3 | 41 | [5.0, 7.4] | [3.0, 3.2] | [10, 15] | [65, 70] |
| LQ4 | 41 | [7.5, 10.9] | [4.2, 4.4] | [15, 20] | [55, 60] |
| LQ5 | 41 | [11, 15] | [4.6, 4.8] | [25, 30] | [45, 50] |

Table 8: Comprehensive degradation parameter selection for 5 levels on RobustSAM dataset and CelebAMask-HQLee et al. (2020).

| Quality score | HQ | LQ1 | LQ2 | LQ3 | LQ4 | LQ5 | avg. |
|---|---|---|---|---|---|---|---|
| CONTRIQUE | 22.06 | 34.81 | 47.41 | 54.69 | 64.68 | 78.48 | 50.36 |
| BRISQUE | 12.97 | 42.45 | 49.27 | 58.40 | 69.36 | 70.56 | 50.50 |

Table 9: Quality scores of different degradation levels, measured by the CONTRIQUE Madhusudana et al. (2022) and the BRISQUE Mittal et al. (2012a) respectively.

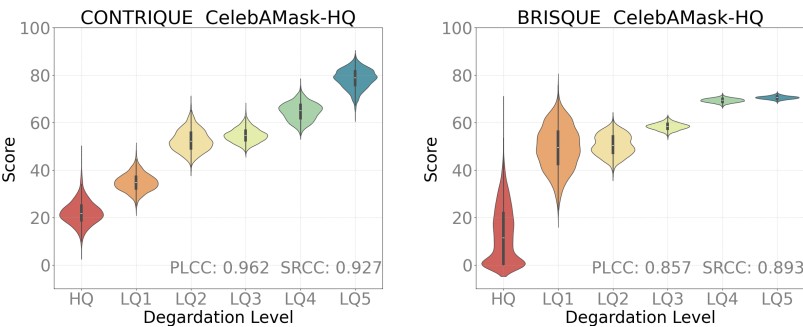

Figure 4: Violin plots of predicted quality scores by IQA models vs. degradation levels, on CelebAMask-HQ.

### A.3 ANALYSIS OF QUAC

#### A.3.1 QUALITY FEATURE

To verify the impact of quality features, we use BRISQUE Mittal et al. (2012a) and CONTRIQUE Madhusudana et al. (2022) as quality functions to implement QuAC. To examine the impact of QuAC on segmentation performance, we first systematically evaluate the effectiveness of the quality function. Specifically, we conducted the following analysis on the CelebAMask-HQ dataset:

- **Quality Assessment:** As shown in Table 9, the quality score of each image was predicted using BRISQUE and CONTRIQUE, respectively.
- **Correlation Analysis:** The Pearson Linear Correlation Coefficient (PLCC) and Spearman Rank Correlation Coefficient (SRCC) were calculated between the predicted scores and the degree of image degradation, with the degree of degradation for high-quality (HQ) images marked as 0.

As shown in Figure 4, both quality functions exhibit significant correlation with image degradation. The predicted score distributions of both methods effectively reflect the gradual changes in degradation levels.

Experimental results show that different quality functions have their own advantages, and the appropriate evaluation metric can be selected based on the specific application scenario. This provides important insights for subsequent research: by combining the advantages of multiple quality features, it is expected to further improve the robustness of the segmentation model under complex degradation conditions.

### A.4 MORE EXPERIMENTS

#### A.4.1 ROBUSTSAM: DETAILED RESULTS

***For 15 types degraded(seen)***, as shown in Table 10 and Table 11, QuAC demonstrates significant advantages under diverse degradation conditions. It shows improvements across all 15 distortion types. Even QuAC$_q$ comprehensively outperforms traditional activation functions. These results demonstrate the necessity of QuAC in complex scenarios. ***For hybrid degradation(unseen)***, as shown in Table 12, Q-ACON demonstrates comprehensive advantages over RobustSAM under unseen hybrid degradation conditions: Q-ACON achieves the best average performance in both mIoU

| mIoU | bright | elastic | color | compress | contrast | fog | frosted | gauss | impulse | ISO | motion | rain | resample | snow | zoom | Avg |
|---|---|---|---|---|---|---|---|---|---|---|---|---|---|---|---|---|
| + ReLU | 89.73 | 81.60 | 89.02 | 78.93 | 88.32 | 89.78 | 86.32 | 88.41 | 88.57 | 86.07 | 81.72 | 88.18 | 87.63 | 84.66 | 80.87 | 85.99 |
| + GELU | 89.66 | 81.77 | 89.17 | 79.70 | 88.50 | 89.71 | 87.23 | 88.93 | 88.66 | 86.47 | 82.04 | 88.49 | 88.24 | 85.17 | 82.91 | 86.44 |
| + SiLU | 89.30 | 82.13 | 88.69 | 80.44 | 88.04 | 89.30 | 87.61 | 88.02 | 88.47 | 85.72 | 81.96 | 88.32 | 87.86 | 85.12 | 82.57 | 86.24 |
| + DY-ReLU-B | 89.33 | 81.96 | 88.84 | 77.99 | 87.96 | 89.67 | 86.94 | 88.22 | 88.17 | 85.80 | 81.85 | 88.13 | 87.84 | 84.34 | 81.38 | 85.89 |
| Q-ReLU$_q$ | 89.47 | 81.69 | 88.99 | 79.44 | 88.28 | 89.46 | 87.55 | 88.60 | 88.60 | 85.64 | 82.03 | 88.45 | 87.99 | 85.00 | 82.53 | 86.22 |
| Q-ReLU | 89.73 | 81.86 | 89.18 | 79.91 | 88.55 | 89.82 | 87.72 | 88.96 | 88.68 | 86.38 | 81.96 | 88.68 | 88.34 | 85.32 | 82.40 | 86.50 |
| + meta-ACON | 89.68 | 81.58 | 89.03 | 79.38 | 88.10 | 89.92 | 87.12 | 88.41 | 88.61 | 85.89 | 81.76 | 88.36 | 87.73 | 84.72 | 81.35 | 86.11 |
| + QuAC$_q$ | 90.17 | 82.20 | 89.51 | 80.62 | 89.05 | 90.27 | 87.82 | 89.34 | 89.09 | 86.90 | 82.46 | 89.29 | 88.65 | 85.88 | 82.58 | 86.92 |
| + QuAC | 90.13 | 82.45 | 89.45 | 81.00 | 88.91 | 90.14 | 88.07 | 89.21 | 89.19 | 86.92 | 82.45 | 89.07 | 88.81 | 85.76 | 83.17 | 86.98 |

Table 10: Segmentation results(mIoU) of RobustSAM model variants on different types of degradation.

| mDice | bright | elastic | color | compress | contrast | fog | frosted | gauss | impulse | ISO | motion | rain | resample | snow | zoom | Avg |
|---|---|---|---|---|---|---|---|---|---|---|---|---|---|---|---|---|
| + ReLU | 94.09 | 88.92 | 93.63 | 87.22 | 93.20 | 94.08 | 92.04 | 93.23 | 93.37 | 91.69 | 89.18 | 93.13 | 92.80 | 90.95 | 88.70 | 91.75 |
| + GELU | 93.99 | 89.04 | 93.67 | 87.72 | 93.27 | 94.04 | 92.56 | 93.54 | 93.36 | 91.95 | 89.38 | 93.31 | 93.14 | 91.31 | 89.95 | 92.02 |
| + SiLU | 93.71 | 89.35 | 93.35 | 88.17 | 92.90 | 93.73 | 92.84 | 92.85 | 93.20 | 91.37 | 89.29 | 93.16 | 92.88 | 91.18 | 89.65 | 91.84 |
| + DY-ReLU-B | 93.79 | 89.19 | 93.49 | 86.46 | 92.94 | 93.96 | 92.40 | 93.11 | 93.08 | 91.49 | 89.29 | 93.09 | 92.94 | 90.71 | 89.00 | 91.66 |
| Q-ReLU$_q$ | 93.82 | 88.99 | 93.55 | 87.38 | 93.04 | 93.80 | 92.74 | 92.93 | 93.28 | 91.29 | 89.34 | 93.26 | 92.93 | 91.13 | 89.68 | 91.81 |
| Q-ReLU | 94.03 | 89.09 | 93.67 | 87.67 | 93.30 | 94.08 | 92.87 | 93.54 | 93.37 | 91.83 | 89.33 | 93.38 | 93.18 | 91.30 | 89.66 | 92.02 |
| + meta-ACON | 94.01 | 88.91 | 93.58 | 87.48 | 92.97 | 94.15 | 92.47 | 93.12 | 93.31 | 91.48 | 89.20 | 93.21 | 92.86 | 90.96 | 89.00 | 91.78 |
| + QuAC$_q$ | 94.35 | 89.44 | 93.94 | 88.42 | 93.67 | 94.40 | 93.03 | 93.82 | 93.69 | 92.22 | 89.74 | 93.87 | 93.49 | 91.76 | 89.83 | 92.38 |
| + QuAC | 94.29 | 89.57 | 93.87 | 88.58 | 93.53 | 94.29 | 93.18 | 93.71 | 93.71 | 92.27 | 89.69 | 93.69 | 93.54 | 91.70 | 90.17 | 92.39 |

Table 11: Segmentation results(mDice) of RobustSAM model variants on different types of degradation.

and mDice. Experimental results demonstrate that Q-ACON effectively improves the model's generalization ability to unseen degradations. Notably, Q-ACON demonstrates excellent robustness to extreme degradations while maintaining high-quality image performance, demonstrating its value in real-world scenarios. It should be noted that we adopt the same mixed distortion method as used in CelebAMask-HQ for the MSRA10K test set. However, since the images in the MSRA10K test set are smaller than those in CelebAMask-HQ, the LQ5 distortion level is almost absent. As a result, our model (QuAC) performs slightly worse than GELU under this setting.

### A.4.2 SEGNEXT: DETAILED RESULTS

Table 13 reports the performance comparison under varying levels of hybrid degradations, along with the overall average results (*avg.*). Q-ACON consistently outperforms other activation functions, demonstrating its effectiveness in enhancing feature representation for degraded images through quality-adaptive activation. In particular, both Q-ACON$_q$ and Q-ACON achieve the highest mIoU scores under severe degradation, while also delivering strong performance across the entire test set. Q-ACON$_q$ shows a slight but consistent advantage over meta-ACON Ma et al. (2021) in handling heavily degraded inputs. These findings highlight the importance of incorporating quality-adaptive mechanisms to improve the robustness of models in challenging visual conditions.

| RobustSAM | mIoU | | | | | | | mDice | | | | | | |
|---|---|---|---|---|---|---|---|---|---|---|---|---|---|---|
| | HQ | LQ1 | LQ2 | LQ3 | LQ4 | LQ5 | avg. | HQ | LQ1 | LQ2 | LQ3 | LQ4 | LQ5 | avg. |
| GELU | 89.56 | 88.67 | 84.82 | 77.31 | 70.96 | 66.16 | 79.58 | 93.95 | 93.44 | 91.13 | 86.20 | 81.75 | 78.17 | 87.44 |
| SiLU | 89.26 | 88.33 | 84.36 | 76.53 | 70.02 | 66.00 | 79.08 | 93.71 | 93.20 | 90.82 | 85.62 | 81.01 | 78.04 | 87.07 |
| ReLU | 89.46 | 88.09 | 84.30 | 76.47 | 69.51 | 62.72 | 78.43 | 93.94 | 93.14 | 90.74 | 85.57 | 80.63 | 75.32 | 86.56 |
| DY-ReLU-B | 88.98 | 88.15 | 84.80 | 76.84 | 70.34 | 64.22 | 78.89 | 93.58 | 93.14 | 91.14 | 85.87 | 81.29 | 76.64 | 86.94 |
| meta-ACON | 89.50 | 88.25 | 84.39 | 76.99 | 70.4 | 65.04 | 79.10 | 93.90 | 93.21 | 90.82 | 85.95 | 81.32 | 77.24 | 87.07 |
| Q-ACON$_q$ | 90.02 | 89.10 | 84.92 | 77.36 | 70.83 | 64.36 | 79.43 | 94.26 | 93.79 | 91.25 | 86.30 | 81.70 | 76.65 | 87.33 |
| Q-ACON | 90.09 | 89.12 | 85.16 | 77.64 | 71.05 | 65.47 | 79.76 | 94.27 | 93.76 | 91.38 | 86.47 | 81.82 | 77.56 | 87.54 |

Table 12: **Face parsing performance** (mIoU and mDice) of RobustSAM with different activation functions on *hybrid degradations (unseen)*.

| SegNeXt | mIoU | | | | | | | mDice | | | | | | |
|---|---|---|---|---|---|---|---|---|---|---|---|---|---|---|
| | HQ | LQ1 | LQ2 | LQ3 | LQ4 | LQ5 | avg. | HQ | LQ1 | LQ2 | LQ3 | LQ4 | LQ5 | avg. |
| ReLU | 71.79 | 68.68 | 63.24 | 59.12 | 55.54 | 51.83 | 61.70 | 81.78 | 78.94 | 74.32 | 70.81 | 67.62 | 64.10 | 72.93 |
| GELU | 72.69 | 68.96 | 63.75 | 59.12 | 55.49 | 51.86 | 61.98 | 82.65 | 79.21 | 74.87 | 70.82 | 67.59 | 64.18 | 73.22 |
| SiLU | **72.87** | 69.29 | **64.01** | 59.74 | 56.17 | 52.29 | 62.40 | 82.70 | 79.54 | 75.12 | 71.46 | 68.32 | 64.67 | 73.64 |
| DY-ReLU-B | 72.56 | 68.47 | 62.86 | 59.01 | 55.58 | 51.92 | 61.73 | 82.56 | 78.80 | 74.05 | 70.67 | 67.61 | 64.16 | 72.98 |
| DiTAC | 72.09 | 68.70 | 63.79 | **60.13** | 56.49 | 52.48 | 62.28 | 82.05 | 78.98 | 74.91 | 71.81 | 68.59 | 64.81 | 73.53 |
| meta-ACON | 72.72 | 69.02 | 63.46 | 59.65 | 56.12 | 51.97 | 62.16 | 82.59 | 79.20 | 74.60 | 71.33 | 68.22 | 64.26 | 73.37 |
| Q-ACON$_q$ | 72.86 | **69.82** | 63.23 | 59.61 | 56.88 | **53.76** | **62.69** | 82.58 | 79.00 | 74.63 | 71.91 | 68.73 | 65.16 | 73.67 |
| Q-ACON | 72.78 | 69.32 | 62.56 | 59.63 | **56.92** | 53.66 | 62.48 | **83.08** | **80.72** | **76.00** | **72.34** | **69.43** | **65.77** | **74.56** |

Table 13: **Semantic segmentation performance** (mIoU and mDice) of SegNeXt with different activation functions on *hybrid degradations (seen)*.

| SegNeXt | HQ | LQ1 | LQ2 | LQ3 | LQ4 | LQ5 | avg. |
|---|---|---|---|---|---|---|---|
| DY-ReLU-B | 72.56 | 68.47 | 62.86 | 59.01 | 55.58 | 51.92 | 61.73 |
| Q-ReLU$_q$ | 72.44 | 68.50 | 63.22 | 59.31 | 55.95 | 52.44 | 61.98 |
| Q-ReLU | **72.95** | **70.06** | **63.67** | **60.60** | **57.73** | **53.99** | **63.17** |
| **RobustSAM** | HQ | LQ1 | LQ2 | LQ3 | LQ4 | LQ5 | avg. |
| DY-ReLU-B | 88.98 | 88.15 | **84.80** | 76.84 | 70.34 | 64.22 | 78.89 |
| Q-ReLU$_q$ | 89.45 | **88.64** | 84.77 | **77.33** | **70.74** | 65.77 | **79.45** |
| Q-ReLU | **89.62** | 88.63 | 84.70 | 76.81 | 70.15 | **65.98** | 79.32 |

Table 14: Performance of DY-RELU-B Chen et al. (2020) and its quality-adaptive version, *i.e.* Q-ReLU.

### A.4.3 QUALITY-ADAPTIVE ReLU (Q-ReLU)

To verify the concept of *Quality-adaptive Activation* (QuAC), we further apply it to DY-RELU-B Chen et al. (2020). We simply concatenate the quality vector $\mathbf{q}$ to the input tensor $\mathbf{x}$ for learning activation parameters in DY-RELU-B. We refer to the quality-adaptive version of ReLU as Q-ReLU. Additionally, we implement a simplified variant of our Q-ReLU by using only the quality tensor $\mathbf{q}$ in the hyperfunction (denoted by Q-ReLU$_q$). As shown in Table 14, Q-ReLU, demonstrates significant advantages over both baselines. In particular, in SegNeXt, Q-ReLU improves the average mIoU for low-quality images (LQ1-LQ5) from 61.73% to 63.17% (**+1.44%**). Notably, this improvement is quality-dependent—the lower the image quality, the more significant the performance gain (**+2.07** percentage points for LQ5). In RobustSAM, Q-ReLUq achieves the best average performance, while Q-ReLU performs best under extremely degraded conditions (LQ5), which shows the effectiveness of QuAC.

### A.4.4 PERFORMANCE WITH FIXED ENCODER

To evaluate the flexibility and effectiveness of QuAC, we conducted face parsing experiments, based on SegNeXt Guo et al. (2022). Specifically, we use the official pre-trained encoder to initialize the model, and fixed the whole encoder during training. Similar to previous experiments, we compare four model variants, i.e. `Official`, `+meta-ACON`, `+QuAC`, and `+QuAC*`. The corresponding results are shown in Table 15.

Obviously, by merely training the decoder and QuAC, with negligible parameters, we can significantly boost the segmentation performance on degraded images. Such comparison results demonstrate the remarkable flexibility and effectiveness of QuAC in adaptively modulating semantic features. Besides, the cost for fine-tuning merely the decoder and QuAC is negligible.

### A.4.5 RADET: DETAILED RESULTS

In our object detection experiments, we select RADet Hai et al. (2023) as our baseline and evaluate it on the Occluded-LINEMOD dataset Brachmann et al. (2014), a widely adopted benchmark for 6D

| *SegNeXt* | **mIoU** | | | | | | |
|---|---|---|---|---|---|---|---|
| ***fixed encoder*** | HQ | LQ1 | LQ2 | LQ3 | LQ4 | LQ5 | avg. |
| Official | 53.92 | 51.12 | 41.54 | 33.46 | 25.94 | 19.87 | 37.64 |
| + meta-ACON | 53.62 | 51.08 | 41.55 | 33.60 | 26.32 | 19.77 | 37.66 |
| + QuAC | 73.79 | 68.79 | 58.75 | **49.62** | **40.93** | **32.73** | **54.10** |
| + QuAC* | **73.90** | **69.17** | **59.01** | 48.56 | 38.12 | 28.81 | 52.93 |

Table 15: Face parsing performance of SegNeXt and its model variants, while the encoder is initialized by the official model and *fixed* during training. In other words, only the prediction head, as well as QuAC, are optimized during training.

| **AP** | HQ | LQ1 | LQ2 | LQ3 | LQ4 | LQ5 | avg. |
|---|---|---|---|---|---|---|---|
| Official | 63.2 | 54.5 | 49.4 | 41.8 | 33.7 | 28.0 | 45.1 |
| + GELU | 63.4 | **54.6** | 48.7 | 41.0 | 33.3 | 27.0 | 44.5 |
| + Q-ReLU | 62.7 | 53.6 | 48.6 | 41.8 | 34.6 | 28.6 | 44.9 |
| + meta-ACON | 63.5 | 53.8 | 49.0 | 42.8 | 35.9 | 29.7 | 45.78 |
| + Q-ACON | 63.6 | 54.4 | **49.8** | **42.9** | **35.5** | **29.4** | **45.9** |

| **AP$_{75}$** | HQ | LQ1 | LQ2 | LQ3 | LQ4 | LQ5 | avg. |
|---|---|---|---|---|---|---|---|
| Official | 74.7 | 63.9 | 57.8 | 46.8 | 35.7 | 28.7 | 51.3 |
| + GELU | **75.1** | **64.0** | 56.7 | 46.5 | 36.0 | 27.9 | 51.0 |
| + Q-ReLU | 74.3 | 63.0 | 56.5 | 47.9 | 37.3 | 29.4 | 51.4 |
| + meta-ACON | 75.0 | 62.7 | 57.0 | 48.7 | 39.2 | 30.7 | 52.2 |
| + Q-ACON | **75.1** | 63.9 | **58.3** | **48.7** | **38.7** | **30.9** | **52.6** |

Table 16: Detection performance (AP and AP$_{75}$) of RADet Hai et al. (2023) model variants, on the Occluded-LINEMOD test set.

pose estimation. This dataset is specifically designed to assess the performance of object detection and pose estimation algorithms under challenging and occluded conditions. We report key evaluation metrics, including AP and AP$_{75}$, which are standard benchmarks in object detection tasks. The results, presented in Table 16, highlight the superior detection capabilities of our proposed QuAC method, particularly in scenarios involving extremely low-quality imagery. Notably, QuAC outperforms other activation functions by achieving significant improvements in both accuracy and robustness.

### A.5 MORE VISUALIZATION RESULTS

As shown in Fig. 5, our proposed QuAC module generates restoration results with superior clarity and more vivid, realistic visual effects compared to the baseline and the meta-ACON variant. Fig. 6 present more face parsing results and general image segmentation results. Both QuAC and QuAC* consistently and lead to better segmentation results, especially near the boundaries. Figure 7 demonstrates performance under hybrid degradation conditions.

### A.6 IMPLEMENTATION DETAILS

In all comparative experiments, we followed the training procedures exactly as described in the original paper, and we have summarized them in Table 17.

- **SegNeXt**: In the face parsing experiment, we choose CelebAMask-HQ Lee et al. (2020) as the dataset, which is split into 24,183 images for training, 2,993 for validation, and 2,824 for testing. The learning rate was set to 6e−5, batch size was set to 8, and the maximum number of iterations was 160K. All experiments were conducted on a single 24GB RTX 4090 GPU.
- **RobustSAM**: For the robustness-aware segmentation experiment, we use the Robust-Seg dataset, consisting of 26,000 training images, 5,229 validation images, and 2,000 test im-

ages. The model was trained for 20 epochs using a learning rate of $1e{-}4$ and a batch size of 2. We utilized two 24GB RTX 4090 GPUs for training.

- **ResNet**: In the low-light scene classification experiment, we adopt the CODaN Lengyel et al. (2021) dataset, which contains 10,000 training images, 500 validation images, and 5,000 test images (2,500 from daytime and 2,500 from nighttime conditions). We evaluate both ResNet18 and ResNet34 architectures, using learning rates of $1e{-}4$ and $1e{-}3$ respectively. Each model was trained for 100 epochs with a batch size of 96 on a single 24GB RTX 4090 GPU.

- **YOLOv11**: For object detection in hazy conditions, we utilize the HazyDet Feng et al. (2024) dataset, which includes 8,000 images for training and 1,000 for validation. The model was trained for 100 epochs with a learning rate of $1e{-}2$ and a batch size of 16. Training was conducted using a single 24GB RTX 4090 GPU.

- **RADet**: In the 6D object pose estimation task under occlusion, we use the Occluded-LINEMOD dataset Brachmann et al. (2014). The model was trained for 30,000 iterations with a learning rate of $1e{-}3$ and a batch size of 16. Experiments were carried out on a single 24GB RTX 4090 GPU. Details on the exact data splits for training, validation, and testing follow the standard Occluded-LINEMOD protocol.

- **DiffUIR**: For the image diffusion reconstruction task, we utilize a merged dataset consisting of 380,250 training samples and 11,404 test samples. The training lasted for 300,000 steps, with a learning rate of $8e{-}5$ and batch size of 10. All training was performed on a single 24GB RTX 4090 GPU.

- **AST**: We compared the performance of QuAC on the raindrop removal task. This dataset, AGAN Qian et al. (2018), has a training set of 861 images and a test set of 58 images. The original paper adopted progressive learning, with the number of iterations and learning rate required for each stage shown in the table. We followed this schedule and used two 24GB RTX 4090 GPUs during training.

- **SinSR**: In the single-image super-resolution task, we adopt a dataset with 26,479 training images patches and 100 test images. The model was trained for 30,000 iterations with a learning rate of $5e{-}5$ and a batch size of 6. Training was done using a single 24GB RTX 4090 GPU.

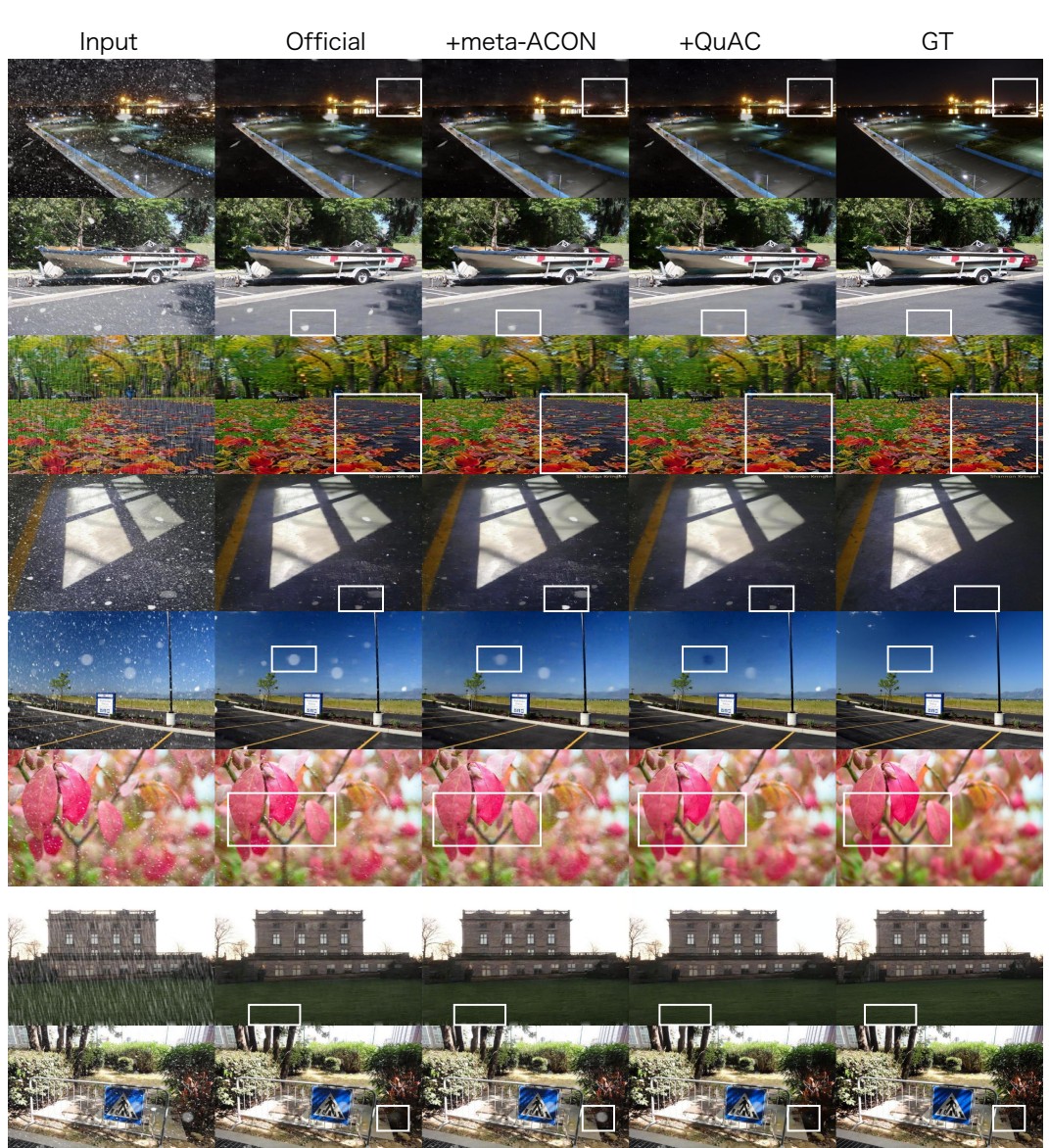

Figure 5: Image restoration results of DiffUIR and its variants. QuAC leads to better visual effects.

| Model | Train dataset | Test dataset | epoch | learning rate | batch size | GPU |
|---|---|---|---|---|---|---|
| SegNeXt | | CelebAMask-HQ (Train:24,183, val:2,993, test:2,824) | 160,000 iterations | 6e-5 | 8 | 24GB RTX4090 GPU |
| RobustSAM | | Robust-Seg (Train:26,000 val:5,229, test:2,000) | 20 | 1e-4 | 2 | 2×24GB RTX4090 GPUs |
| ResNet | | CODaN (Train:10,000, val:500, test:2,500 daytime and 2,500 nighttime) | 100 | ResNet18:1e-4 ResNet34:1e-3 | 96 | 24GB RTX4090 GPU |
| YOLOv11 | | HazyDet (Train:8,000, val:1,000) | 100 | 1e-2 | 16 | 24GB RTX4090 GPU |
| RADet | | Occluded-LINEMOD | 100,000 iterations | 1e-3 | 16 | 24GB RTX4090 GPU |
| DiffUIR | | Merged Dataset (Train:380,250, test:11,404) | 300,000 steps | 8e-5 | 10 | 24GB RTX4090 GPU |
| AST | | AGAN (Train:861 Test:58) | Stage1:300 Stage2:200 Stage3:150 | Stage1:2e-4 Stage2:1e-4 Stage3:8e-5 | Stage1:32 Stage2:10 Stage3:4 | 2×24GB RTX4090 GPUs |
| SinSR | | Dataset (Train:26,479, test:100) | 30,000 iterations | 5e-5 | 6 | 24GB RTX4090 GPU |

Table 17: Experiments details in various tasks.

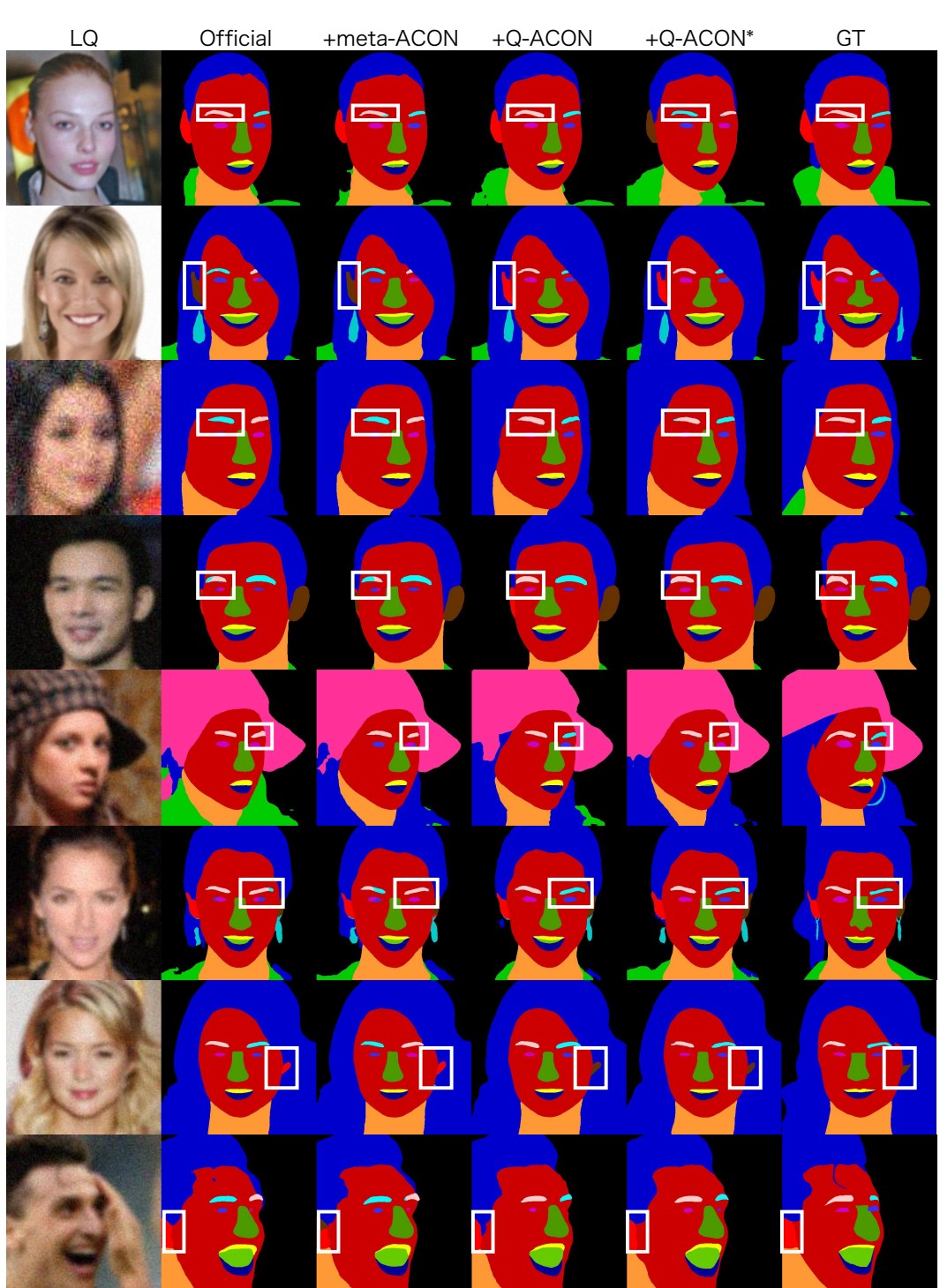

Figure 6: Face parsing results of SegNeXt and its variants. QuAC and QuAC* lead to better results.

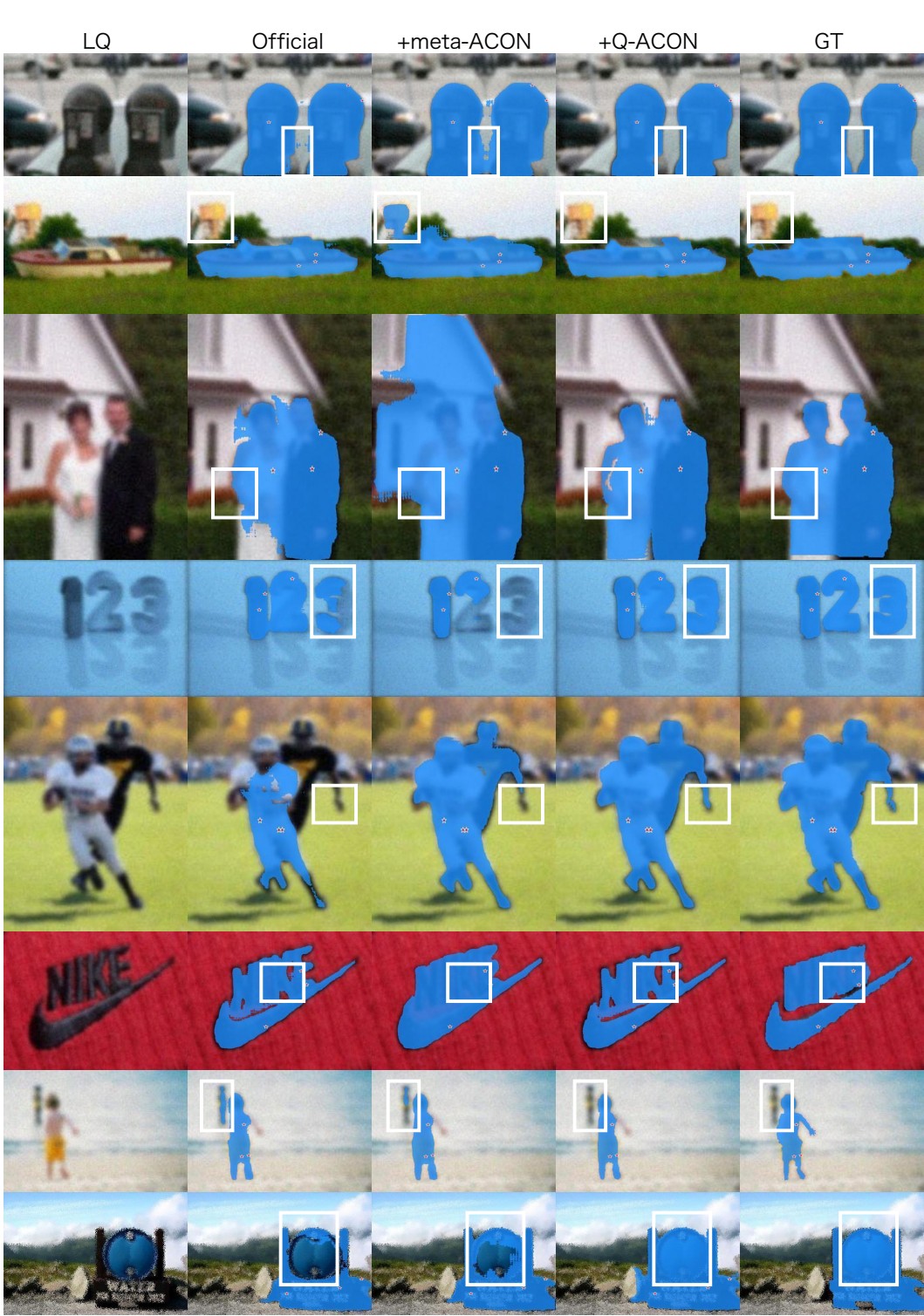

Figure 7: Segmentation results of RobustSAM and its variant with QuAC. QuAC leads to better qualitative results.

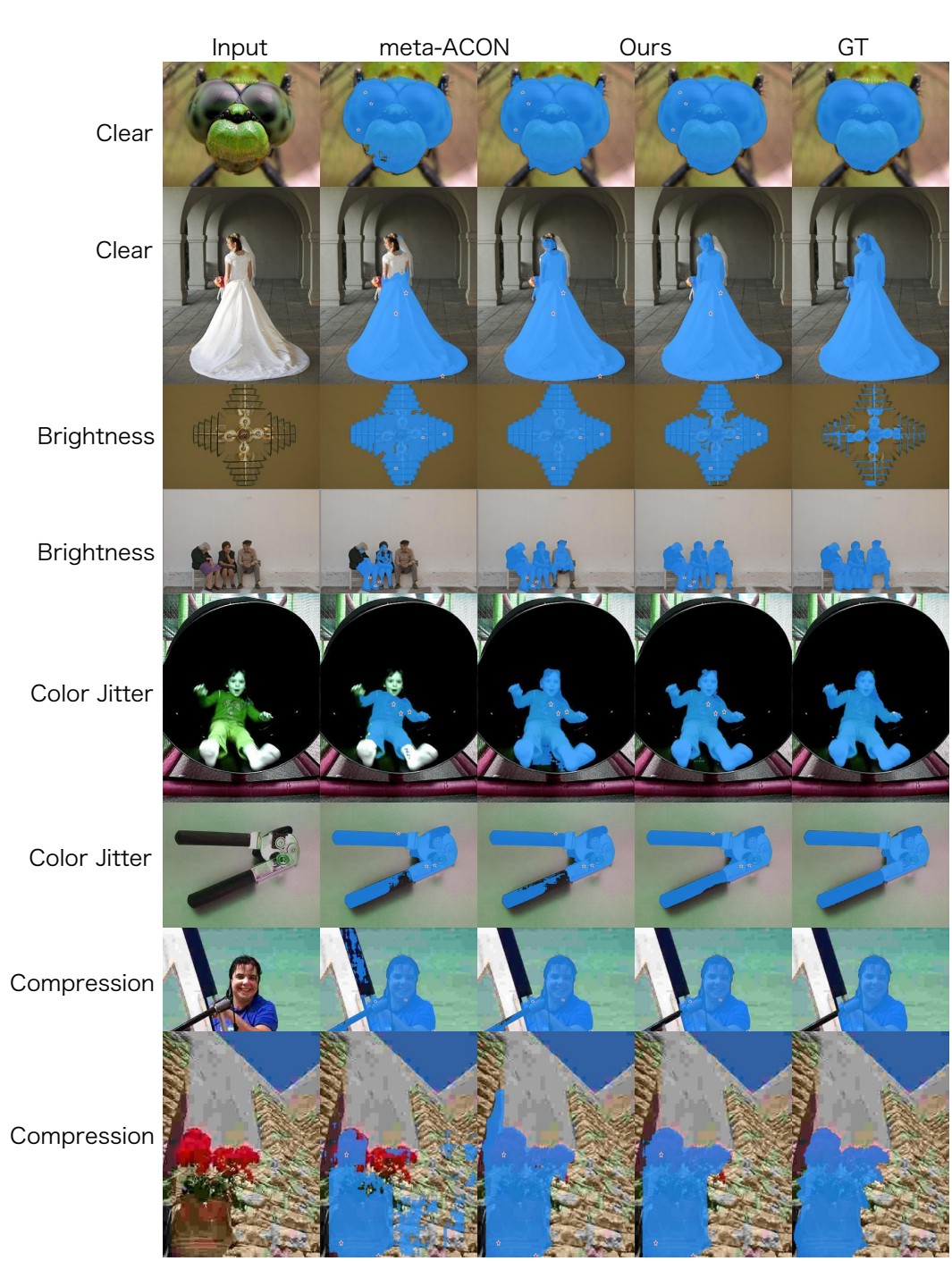

Figure 8: Segmentation results of RobustSAM and its variant with QuAC. QuAC leads to better qualitative results.

