# OpenReview forum: "QuAC: Quality-Adaptive Activation for Degraded Image Understanding"
_ICLR.cc/2026/Conference — ICLR 2026 Conference Withdrawn Submission_

### Official Review · Reviewer_4rBy · 2025-10-27

**Soundness:** 2
**Presentation:** 3
**Contribution:** 2
**Rating:** 4
**Confidence:** 4

**Summary:**

This paper is targeted at image understanding task when the image is of low-quality. The authors propose to insert one quality-adaptive activation module into the backbone model, which can adjust the model activation based on the image quality. In this way, the model could be adapted to image degradations while few parameters are altered. Experiments on various image understanding tasks show the effectiveness of the method.

**Strengths:**

1. The proposed method can be adopted by different image understanding tasks. This shows the generalization of the method.

2. Instead of concatenating one image restoration module before the backbone, the proposed method is more effective through coping with the feature of the backbone model.

3. The writing is easy-to-follow.

**Weaknesses:**

1. The proposed method is still one implementation of hypernetwork so that the comparison between "dynamic activation" and "quality-adaptive activation" in Figure 1(b) is not reasonable.

2. Rectifying the feature of model to deal with image degradation has already been proposed previously in [1],[2] and their methods are highly similar to this paper.

3. The degradation types are pretty limited in experiments. Only blurring, noise, JPEG compression are considered, which are not enough to represent most conditions in reality.

4. The comparison is not sufficient. The method is only compared with backbone with different activations. There should be comparison with models when concatenating image restoration models before the backbone.

[1] Wang, Yang, et al. "Deep degradation prior for low-quality image classification." Proceedings of the IEEE/CVF Conference on Computer Vision and Pattern Recognition. 2020.

[2]Zhou, Shengchao, et al. "Robust feature rectification of pretrained vision models for object recognition." Proceedings of the AAAI Conference on Artificial Intelligence. Vol. 37. No. 3. 2023.

**Questions:**

1. Since a new module is plugged into the backbone network, will the performance on high-quality images be affected?

---

### Official Review · Reviewer_CeJy · 2025-10-27

**Soundness:** 2
**Presentation:** 3
**Contribution:** 2
**Rating:** 4
**Confidence:** 4

**Summary:**

The paper proposes QuAC, a quality-adaptive activation mechanism that modulates neuron activations using image-quality cues to reduce the HQ-LQ domain gap. An instantiation, Q-ACON, plugs into existing models with minimal overhead and yields consistent gains across segmentation, detection, classification, and restoration, including under unseen degradations and real-world conditions.

**Strengths:**

1. Adapting activations via BIQA-derived quality representations is a simple, plug-and-play idea applicable to multiple activation families and architectures (CNNs, transformers, diffusion), with clear formalization and minimal architectural disruption.
2. Broad experiments across tasks/datasets plus ablations show consistent improvements with small parameter cost.

**Weaknesses:**

1. While the paper reports extensive experiments, the motivation needs to be articulated more clearly, particularly the rationale for introducing a quality-perception module.
2. Different tasks often require different architectures (e.g., super-resolution may favor diffusion-based networks). These design choices should be taken into account; the current implementation does not feel simple or elegant.
3. The related-work section omits several advanced BIQA methods, such as VLM-based Q-Instruct and VisualQuality-R1.
4. The approach fails when a corresponding high-quality reference image is unavailable.

**Questions:**

See weaknesses.

---

### Official Review · Reviewer_4KCV · 2025-11-01

**Soundness:** 4
**Presentation:** 4
**Contribution:** 4
**Rating:** 4
**Confidence:** 3

**Summary:**

The paper introduces QuAC (Quality-Adaptive Activation), an innovative approach to improving the performance of image understanding tasks under degraded conditions. Inspired by human visual system mechanisms, QuAC dynamically adjusts neuron activations based on the input image's quality, enhancing the semantic representations for degraded images. The approach is implemented using Quality-Adaptive Meta-ACON (Q-ACON), a flexible, efficient, and plug-and-play method that integrates seamlessly with convolutional networks, transformers, and diffusion models. Extensive experiments demonstrate that QuAC consistently improves the performance of various vision tasks, including segmentation, detection, classification, and restoration, especially in the presence of complex image degradations.

**Strengths:**

1.	Innovative Approach: The paper introduces the novel concept of Quality-adaptive Activation (QuAC), which dynamically adjusts activation functions based on the quality of the input image. This is a unique and meaningful contribution to improving image understanding tasks under degraded conditions.
2.	Wide Applicability: QuAC is shown to be effective across different neural network architectures such as CNNs, transformers, and diffusion models, demonstrating its versatility in multiple vision tasks like segmentation, detection, and restoration.
3.	Performance Improvement: Experimental results show that QuAC, specifically the quality-adaptive meta-ACON (Q-ACON), significantly enhances model performance, particularly in challenging low-quality image conditions, offering improvements over traditional activation functions.

**Weaknesses:**

1. Limited Evaluation on Real-World Datasets: The paper primarily focuses on synthetic degradations and might benefit from more comprehensive evaluation on diverse real-world datasets to demonstrate its robustness and generalizability beyond controlled experiments.
2， Hyperparameter Sensitivity: The dependency on quality assessment models (e.g., BRISQUE) for the quality tensor may raise concerns about the sensitivity of the method to these models. It would be helpful to explore the impact of different quality assessment models (Like LIQE, CLIP-IQA, Q-Align, DpeictQA and some Full-reference IQA metric: LPIPS, etc) on the performance of QuAC.

**Questions:**

please see weakness.

---

### Official Review · Reviewer_hPif · 2025-11-03

**Soundness:** 3
**Presentation:** 2
**Contribution:** 3
**Rating:** 6
**Confidence:** 3

**Summary:**

The authors develop a framework for incorporating perceptual quality into a network, so as to improve performance under input degradation. Specifically, quality information is used to gate activations using the ACON activation function of Ma et al., 2021.

Extensive experiments show modest but consistent performance improvements on a diverse set of networks and applications (segmentation, detection, restoration, classification, etc).

**Strengths:**

Robustness to degradations is important.
The method is simple, and tested extensively.

**Weaknesses:**

The paper is dense in details, and some aspects of the text and figures are not clear.

Method: please define the function sigma (eq. 3), which is critical part of the method.

- For results in table 1, and elsewhere, are the networks (including Q-ACON)  trained jointly on both conditions (Clear / Degrade), or are they trained separately for each?  Also, can you compute standard deveations for the reported values?  Without them, it is not possibe to know how significant the differences between methods are.

- Fig 2:  This figure shows both the archecture and pipeline for computing QuAC, as well as the training setup.   caption should provide more information (top box expresses a QuaC-enabled application (e.g., segmentation).  Bottom left box computes training signals for high-quality images. etc.

- Fig 3: It is difficult to see much difference between the images arising from different methods.  Provide indications in the text for what readers should look for.  Same for figure A5.

**Questions:**

See above.

---

### Note · Authors · 2025-12-01

I have read and agree with the venue's withdrawal policy on behalf of myself and my co-authors.